# EXPLICIT INDUCTION BIAS FOR TRANSFER LEARNING WITH CONVOLUTIONAL NETWORKS

## ABSTRACT

In inductive transfer learning, fine-tuning pre-trained convolutional networks substantially outperforms training from scratch. When using fine-tuning, the underlying assumption is that the pre-trained model extracts generic features, which are at least partially relevant for solving the target task, but would be difficult to extract from the limited amount of data available on the target task. However, besides the initialization with the pre-trained model and the early stopping, there is no mechanism in fine-tuning for retaining the features learned on the source task. In this paper, we investigate several regularization schemes that explicitly promote the similarity of the final solution with the initial model. We eventually recommend a simple $L^2$ penalty using the pre-trained model as a reference, and we show that this approach behaves much better than the standard scheme using weight decay on a partially frozen network.

## 1 INTRODUCTION

It is now well known that modern convolutional neural networks (e.g. Krizhevsky et al. 2012, Simonyan & Zisserman 2015, He et al. 2016, Szegedy et al. 2016) can achieve remarkable performance on large-scale image databases, e.g. ImageNet (Deng et al. 2009) and Places 365 (Zhou et al. 2017), but it is really dissatisfying to see the vast amounts of data, computing time and power consumption that are necessary to train deep networks. Fortunately, such convolutional networks, once trained on a large database, can be refined to solve related but different visual tasks by means of transfer learning, using fine-tuning (Yosinski et al. 2014, Simonyan & Zisserman 2015).

Some form of knowledge is believed to be extracted by learning from the large-scale database of the source task and this knowledge is then transferred to the target task by initializing the network with the pre-trained parameters. However, after fine-tuning, some of the parameters may be quite different from their initial values, resulting in possible losses of general knowledge that may be relevant for the targeted problem. In particular, during fine-tuning, $L^2$ regularization drives the parameters towards the origin and thereby encourages large deviations between the parameters and their initial values.

In order to help preserve the acquired knowledge embedded in the initial network, we consider using other parameter regularization methods during fine-tuning. We argue that the standard $L^2$ regularization, which drives the parameters towards the origin, is not adequate in the framework of transfer learning where the initial values provide a more sensible reference point than the origin. This simple modification keeps the original control of overfitting, by constraining the effective search space around the initial solution, while encouraging committing to the acquired knowledge. We show that it has noticeable effects in inductive transfer learning scenarios.

This paper aims at improving transfer learning by requiring less labeled training data. A form of transfer learning is thus considered, where some pieces of knowledge, acquired when solving a previous learning problem, have to be conveyed to another learning problem. Under this setting, we explore several parameter regularization methods that can explicitly retain the knowledge acquired on the source problem. We investigate variants of $L^2$ penalties using the pre-trained model as reference, which we name $L^2$-*SP* because the pre-trained parameters represent the starting point (-*SP*) of the fine-tuning process. In addition, we evaluate other regularizers based on the Lasso and Group-Lasso penalties, which can freeze some individual parameters or groups of parameters to the pre-trained parameters. Fisher information is also taken into account when we test $L^2$-*SP*

and Group-Lasso-*SP* approaches. Our experiments indicate that all tested parameter regularization methods using the pre-trained parameters as a reference get an edge over the standard $L^2$ weight decay approach. We also analyze the effect of $L^2$-*SP* with theoretical arguments and experimental evidence to recommend using $L^2$-*SP* for transfer learning tasks.

## 2 RELATED WORK

In this section, we recall the existing works dedicated to inductive transfer learning in convolutional networks. We focus on approaches based on the kind of parameter regularization techniques we advocate here. We also recall the existence of similar regularization techniques that were previously applied to different models, especially support vector machines (SVM).

We follow here the nomenclature of Pan & Yang (2010), who categorized several types of transfer learning. A domain corresponds to the feature space and its distribution, whereas a task corresponds to the label space and its conditional distribution with respect to features. The initial learning problem is defined on the source domain and source task, whereas the new learning problem is defined on the target domain and the target task.

In the typology of Pan & Yang, we consider the inductive transfer learning setting, where the target domain is identical to the source domain, and the target task is different from the source task. We furthermore focus on the case where a vast amount of data was available for training on the source problem, and some limited amount of labeled data is available for solving the target problem. Under this setting, we aim at improving the performance on the target problem through parameter regularization methods that explicitly encourage the similarity of the solutions to the target and source problems. We refer here to works on new problems that were formalized or popularized after (Pan & Yang 2010), such as continual learning or fine-tuning, but Pan & Yang's typology remains valid.

**Inductive Transfer Learning for Deep Networks** Donahue et al. (2014) repurposed features extracted from different layers of the pre-trained AlexNet of Krizhevsky et al. (2012) and plugged them into an SVM or a logistic regression classifier. This approach outperformed the state of the art of that time on the Caltech-101 database (Fei-Fei et al. 2006). Later, Yosinski et al. (2014) showed that fine-tuning the whole AlexNet resulted in better performances than using the network as a static feature extractor. Fine-tuning pre-trained VGG (Simonyan & Zisserman 2015) on the image classification task of VOC-2012 (Everingham et al. 2010) and Caltech 256 (Griffin et al. 2007) achieved the best results of that time. Ge & Yu (2017) proposed a scheme for selecting a subset of images from the source problem that have similar local features to those in the target problem and then jointly fine-tuned a pre-trained convolutional network to improve performance on the target task. Besides image classification, many procedures for object detection (Girshick et al. 2014, Redmon et al. 2016, Ren et al. 2015) and image segmentation (Long et al. 2015a, Chen et al. 2017, Zhao et al. 2017) have been proposed relying on fine-tuning to improve over training from scratch. These approaches showed promising results in a challenging transfer learning setup, as going from classification to object detection or image segmentation requires rather heavy modifications of the architecture of the network.

The success of transfer learning with convolutional networks relies on the generality of the learned representations that have been constructed from a large database like ImageNet. Yosinski et al. (2014) also quantified the transferability of these pieces of information in different layers, e.g. the first layers learn general features, the middle layers learn high-level semantic features and the last layers learn the features that are very specific to a particular task. That can be also noticed by the visualization of features (Zeiler & Fergus 2014). Overall, the learned representations can be conveyed to related but different domains and the parameters in the network are reusable for different tasks.

**Parameter Regularization** Parameter regularization can take different forms in deep learning. $L^2$ regularization has been used for a long time as a very simple method for preventing overfitting by penalizing the $L^2$ norm of the parameter vector. It is the usual regularization used in deep learning, including for fine-tuning. $L^1$ prevents overfitting by zeroing out some weights. Max-norm regularization (Srebro & Shraibman 2005) is a hard constraint on the $L^2$ norm that was found especially

helpful when using dropout (Srivastava et al. 2014). Xie et al. (2017) proposed the orthonormal regularizer that encourages orthonormality between all the kernels belonging to the same layer.

In lifelong learning (Thrun & Mitchell 1995, Pentina & Lampert 2015) where a series of tasks is learned sequentially by a single model, the knowledge extracted from the previous tasks may be lost as new tasks are learned, resulting in what is known as catastrophic forgetting. In order to achieve a good performance on all tasks, Li & Hoiem (2017) proposed to use the outputs of the target examples, computed by the original network on the source task, to define a learning scheme preserving the memory of the source tasks when training on the target task. They also tried to preserve the pre-trained parameters instead of the outputs of examples but they did not obtain interesting results. Kirkpatrick et al. (2017) developed a similar approach with success, getting sensible improvements by measuring the sensitivity of the parameters of the network learned on the source data thanks to the Fisher information. This measure is used as the metric in their regularization scheme, elastic weight consolidation, in order to preserve the representation learned on the source data, which is required to retain the solutions learned on the previous tasks. In their experiments, elastic weight consolidation was shown to avoid forgetting, but fine-tuning with plain stochastic gradient descent was more effective than elastic weight consolidation for learning new tasks. Hence, elastic weight consolidation may be thought as being inadequate for transfer learning, where performance is only measured on the target task. We will show that this conclusion is not appropriate in typical transfer learning scenarios.

In domain adaptation (Long et al. 2015b, Tzeng et al. 2016), where the target task is identical to the source task and no (or a small quantity of) target data is labeled, most approaches are searching for a common representation space for source and target domains to reduce domain shift. Rozantsev et al. (2016) introduced a parameter regularization for keeping the similarity between the pre-trained model and the fine-tuned model. Since domain adaptation needs more effort on reducing domain shift, their regularization was more flexible with the exponential function of a linear transformation. We found in our preliminary experiments that the exponential term was able to improve the results but not as much as $L^2$-*SP*. The gradient of the exponential term indicates that when the weight goes farther, the force for bringing it back is exponentially stronger.

Regularization has been a means to build shrinkage estimators for decades. Shrinking towards zero is the most common form of shrinkage, but shrinking towards adaptively chosen targets has been around for some time, starting with Stein shrinkage (see e.g. Lehmann & Casella 1998, chapter 5), and more recently with SVM. For example, Yang et al. (2007) proposed an adaptive SVM (A-SVM), which regularizes the squared difference between the parameter vector and an initial parameter vector that is learned from the source database. Then, Aytar & Zisserman (2011) added a linear relaxation to A-SVM and proposed the projective model transfer SVM (PMT-SVM), which regularizes the angle between the parameter vector and the initial one. Experiments in Aytar & Zisserman (2011), Tommasi et al. (2014) demonstrated that both A-SVM and PMT-SVM were able to outperform standard $L^2$ regularization with limited labeled data in the target task. These relatives differ from the present proposal in two respects. Technically, they were devised for binary classification problems, even if multi-class classification can be addressed by elaborate designs. More importantly, they consider a fixed representation, and transfer aims at learning similar classification parameters in that space. Here, with deep networks, transfer aims at learning similar representations upon which classification parameters will be learned from scratch. Hence, even though the techniques we propose here are very similar regarding regularization functions, they operate on very different objects. Thus, to the best of our knowledge, we present the first results on transfer learning with convolutional networks that are based on the regularization terms described in the following section.

## 3 REGULARIZERS FOR FINE-TUNING

In this section, we detail the penalties we consider for fine-tuning. Parameter regularization is critical when learning from small databases. When learning from scratch, regularization is aimed at facilitating optimization and avoiding overfitting, by implicitly restricting the capacity of the network, that is, the effective size of the search space. In transfer learning, the role of regularization is similar, but the starting point of the fine-tuning process conveys information that pertain to the source problem (domain and task). Hence, the network capacity has not to be restricted blindly: the

pre-trained model sets a reference that can be used to define the functional space effectively explored during fine-tuning.

Since we are using early stopping, fine-tuning a pre-trained model is an implicit form of induction bias towards the initial solution. We explore here how a coherent explicit induction bias, encoded by a regularization term, affects the training process. Section 4 shows that all such schemes get an edge over the standard approaches that either use weight decay or freeze part of the network for preserving the low-level representations that are built in the first layers of the network.

Let $\boldsymbol{w} \in \mathbb{R}^n$ be the parameter vector containing all the network parameters that are to be adapted to the target task. The regularized objective function $\tilde{J}$ that is to be optimized is the sum of the standard objective function $J$ and the regularizer $\Omega(\boldsymbol{w})$. In our experiments, $J$ is the negative log-likelihood, so that the criterion $\tilde{J}$ could be interpreted in terms of maximum *a posteriori* estimation, where the regularizer $\Omega(\boldsymbol{w})$ would act as the log prior of $\boldsymbol{w}$. More generally, the minimizer of $\tilde{J}$ is a trade-off between the data-fitting term and the regularization term.

**$L^2$ penalty**   Our baseline penalty for transfer learning is the usual $L^2$ penalty, also known as weight decay, since it drives the weights of the network to zero:

$$\Omega(\boldsymbol{w}) = \frac{\alpha}{2} \left\| \boldsymbol{w} \right\|_2^2 \quad, \tag{1}$$

where $\alpha$ is the regularization parameter setting the strength of the penalty and $\left\| \cdot \right\|_p$ is the $p$-norm of a vector.

**$L^2$-SP**   Let $\boldsymbol{w}^0$ be the parameter vector of the model pre-trained on the source problem, acting as the starting point (-*SP*) in fine-tuning. Using this initial vector as the reference in the $L^2$ penalty, we get:

$$\Omega(\boldsymbol{w}) = \frac{\alpha}{2} \left\| \boldsymbol{w} - \boldsymbol{w}^0 \right\|_2^2 \quad. \tag{2}$$

Typically, the transfer to a target task requires slight modifications of the network architecture used for the source task, such as on the last layer used for predicting the outputs. Then, there is no one-to-one mapping between $\boldsymbol{w}$ and $\boldsymbol{w}^0$, and we use two penalties: one for the part of the target network that shares the architecture of the source network, denoted $\boldsymbol{w}_{\mathcal{S}}$, the other one for the novel part, denoted $\boldsymbol{w}_{\bar{\mathcal{S}}}$. The compound penalty then becomes:

$$\Omega(\boldsymbol{w}) = \frac{\alpha}{2} \left\| \boldsymbol{w}_{\mathcal{S}} - \boldsymbol{w}_{\mathcal{S}}^0 \right\|_2^2 + \frac{\beta}{2} \left\| \boldsymbol{w}_{\bar{\mathcal{S}}} \right\|_2^2 \quad. \tag{3}$$

**$L^2$-SP-Fisher**   Elastic weight consolidation (Kirkpatrick et al. 2017) was proposed to avoid catastrophic forgetting in the setup of lifelong learning, where several tasks should be learned sequentially. In addition to preserving the initial parameter vector $\boldsymbol{w}^0$, it consists in using the estimated Fisher information to define the distance between $\boldsymbol{w}_{\mathcal{S}}$ and $\boldsymbol{w}_{\mathcal{S}}^0$. More precisely, it relies on the diagonal of the Fisher information matrix, resulting in the following penalty:

$$\Omega(\boldsymbol{w}) = \frac{\alpha}{2} \sum_{j \in \mathcal{S}} \hat{F}_{jj} \left( w_j - w_j^0 \right)^2 + \frac{\beta}{2} \left\| \boldsymbol{w}_{\bar{\mathcal{S}}} \right\|_2^2 \quad, \tag{4}$$

where $\hat{F}_{jj}$ is the estimate of the $j$th diagonal element of the Fisher information matrix. It is computed as the average of the squared Fisher's score on the source problem, using the inputs of the source data:

$$\hat{F}_{jj} = \frac{1}{m} \sum_{i=1}^{m} \sum_{k=1}^{K} f_k(\boldsymbol{x}^{(i)}; \boldsymbol{w}^0) \left( \frac{\partial}{\partial w_j} \log f_k(\boldsymbol{x}^{(i)}; \boldsymbol{w}^0) \right)^2 \quad,$$

where the outer average estimates the expectation with respect to inputs $\boldsymbol{x}$ and the inner weighted sum is the estimate of the conditional expectation of outputs given input $\boldsymbol{x}^{(i)}$, with outputs drawn from a categorical distribution of parameters $(f_1(\boldsymbol{x}^{(i)}; \boldsymbol{w}), \ldots, f_k(\boldsymbol{x}^{(i)}; \boldsymbol{w}), \ldots, f_K(\boldsymbol{x}^{(i)}; \boldsymbol{w}))$.

**$L^1$-SP**  We also experiment the $L^1$ variant of $L^2$-SP:

$$\Omega(\boldsymbol{w}) = \alpha \left\| \boldsymbol{w}_{\mathcal{S}} - \boldsymbol{w}_{\mathcal{S}}^0 \right\|_1 + \frac{\beta}{2} \left\| \boldsymbol{w}_{\bar{\mathcal{S}}} \right\|_2^2 \quad . \tag{5}$$

The usual $L^1$ penalty encourages sparsity; here, by using $\boldsymbol{w}_{\mathcal{S}}^0$ as a reference in the penalty, $L^1$-SP encourages some components of the parameter vector to be frozen, equal to the pre-trained initial values. The penalty can thus be thought as intermediate between $L^2$-SP (3) and the strategies consisting in freezing a part of the initial network. We explore below other ways of doing so.

**Group-Lasso-*SP* (*GL-SP*)**  Instead of freezing some individual parameters, we may encourage freezing some groups of parameters corresponding to channels of convolution kernels. Formally, we endow the set of parameters with a group structure, defined by a fixed partition of the index set $\mathcal{I} = \{1, \dots, p\}$, that is, $\mathcal{I} = \bigcup_{g=0}^{G} \mathcal{G}_g$, with $\mathcal{G}_g \cap \mathcal{G}_h = \emptyset$ for $g \neq h$. In our setup, $\mathcal{G}_0 = \bar{\mathcal{S}}$, and for $g > 0$, $\mathcal{G}_g$ is the set of fan-in parameters of channel $g$. Let $p_g$ denote the cardinality of group $g$, and $\boldsymbol{w}_{\mathcal{G}_g} \in \mathbb{R}^{p_g}$ be the vector $(w_j)_{j \in \mathcal{G}_g}$. Then, the *GL-SP* penalty is:

$$\Omega(\boldsymbol{w}) = \frac{\beta}{2} \left\| \boldsymbol{w}_{\mathcal{G}_0} \right\|_2^2 + \alpha \sum_{g=1}^{G} \alpha_g \left\| \boldsymbol{w}_{\mathcal{G}_g} - \boldsymbol{w}_{\mathcal{G}_g}^0 \right\|_2 \quad , \tag{6}$$

where $\boldsymbol{w}_{\mathcal{G}_0}^0 = \boldsymbol{w}_{\bar{\mathcal{S}}}^0 \overset{\triangle}{=} \boldsymbol{0}$, and, for $g > 0$, $\alpha_g$ is a predefined constant that may be used to balance the different cardinalities of groups. In our experiments, we used $\alpha_g = p_g^{1/2}$.

Our implementation of Group-Lasso-*SP* can freeze feature extractors at any depth of the convolutional network, to preserve the pre-trained feature extractors as a whole instead of isolated pretrained parameters. The group $\mathcal{G}_g$ of size $p_g = h_g \times \mathrm{w}_g \times d_g$ gathers all the parameters of a convolution kernel of height $h_g$, width $\mathrm{w}_g$, and depth $d_g$. This grouping is done at each layer of the network, for each output channel, so that the group index $g$ corresponds to two indexes in the network architecture: the layer index $l$ and the output channel index at layer $l$. If we have $c_l$ such channels at layer $l$, we have a total of $G = \sum_l c_l$ groups.

**Group-Lasso-SP-Fisher (*GL-SP-Fisher*)**  Following the idea of $L^2$-*SP-Fisher*, the Fisher version of *GL-SP* is:

$$\Omega(\boldsymbol{w}) = \frac{\beta}{2} \left\| \boldsymbol{w}_{\mathcal{G}_0} \right\|_2^2 + \alpha \sum_{g=1}^{G} \alpha_g \left( \sum_{j \in \mathcal{G}_g} \hat{F}_{jj} \left( w_j - w_j^0 \right)^2 \right)^{1/2} \quad . \tag{7}$$

## 4 EXPERIMENTS

We evaluate the aforementioned parameter regularizers on several pairs of source and target tasks. We use ResNet (He et al. 2016) as our base network, since it has proven its wide applicability on transfer learning tasks. Conventionally, if the target task is also a classification task, the training process starts by replacing the last layer with a new one, randomly generated, whose size depends on the number of classes in the target task. All mentioned parameter regularization approaches are applied to all layers except new layers, and parameters in new layers are regularized by $L^2$ penalty as described in Section 3.

### 4.1 SOURCE AND TARGET DATABASES

For comparing the effect of similarity between the source problem and the target problem on transfer learning, we have chosen two source databases: ImageNet (Deng et al. 2009) for generic object recognition and Places 365 (Zhou et al. 2017) for scene classification. Likewise, we have three different databases related to three target problems: Caltech 256 (Griffin et al. 2007) contains different objects for generic object recognition, similar to ImageNet; Stanford Dogs 120 (Khosla et al. 2011) contains images of 120 breeds of dogs; MIT Indoors 67 (Quattoni & Torralba 2009) consists of 67 indoor scene categories.

Table 1: Characteristics of the target databases: name and type, numbers of training and test images per class, and number of classes.

| Database | task category | # training images | # test images | # classes |
|---|---|---|---|---|
| Caltech 256 – 30 | generic object recog. | 30 | > 20 | 257 |
| Caltech 256 – 60 | generic object recog. | 60 | > 20 | 257 |
| Stanford Dogs 120 | specific object recog. | 100 | ∼ 72 | 120 |
| MIT Indoors 67 | scene classification | 80 | 20 | 67 |

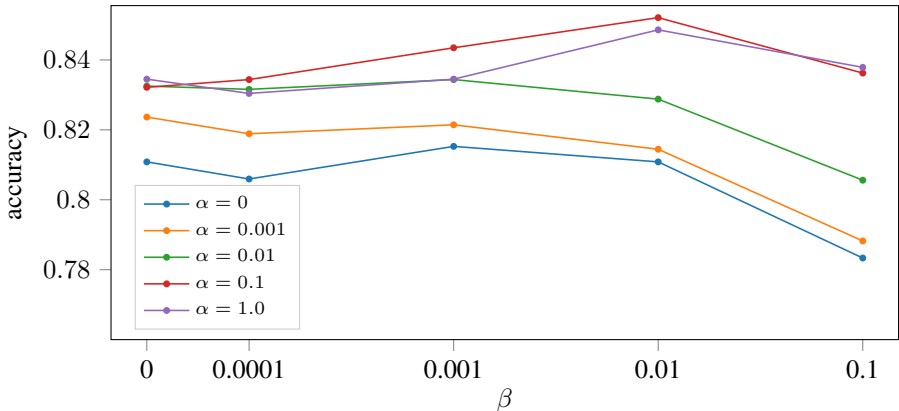

Figure 1: Accuracy on Stanford Dogs 120 for $L^2$-SP, according to the two regularization hyper-parameters $\alpha$ and $\beta$ respecively applied to the layers inherited from the source task and the last classification layer (see Equation 3).

Each target database is split into training and testing sets following the suggestion of their creators. We consider two configurations for Caltech 256: 30 or 60 examples randomly drawn from each category for training, using the remaining examples for test. Stanford Dogs 120 has exactly 100 examples per category for training and 8580 examples in total for testing. As for MIT Indoors 67, there are exactly 80 examples per category for training and 20 examples per category for testing. See Table 1 for details.

## 4.2 TRAINING DETAILS

Most images in those databases are color images. If not, we create a three-channel image by duplicating the gray-scale data. All images are pre-processed: we resize images to 256×256 and subtract the mean activity computed over the training set from each channel, then we adopt random blur, random mirror and random crop to 224×224 for data augmentation. The network parameters are regularized as described in Section 3. Cross validation is used for searching the best regularization hyperparameters $\alpha$ and $\beta$: $\alpha$ differs across experiments, and $\beta = 0.01$ is consistently picked by cross-validation for regularizing the last layer. Figure 1 illustrates that the test accuracy varies smoothly according to the regularization strength, and that there is a sensible benefit in penalizing the last layer. When applicable, the Fisher information matrix is estimated on the source database. The two source databases (ImageNet or Places 365) yield different estimates. Regarding testing, we use central crops as inputs to compute the classification accuracy.

Stochastic gradient descent with momentum 0.9 is used for optimization. We run 9000 iterations and divide the learning rate by 10 after 6000 iterations. The initial learning rates are 0.005, 0.01 or 0.02, depending on the tasks. Batch size is 64. Then under the best configuration, we repeat five times the learning process to obtain an average classification precision and standard deviation. All the experiments are performed with Tensorflow (Abadi et al. 2015).

Table 2: Average classification accuracies of $L^2$, $L^2$-SP and $L^2$-SP-Fisher on 5 different runs. The first line is the reference of selective joint fine-tuning (Ge & Yu 2017) that selects 200,000 images from the source database during transfer learning.

|  | MIT Indoors 67[1] | Stanford Dogs 120 | Caltech 256 − 30 | Caltech 256 − 60 |
|---|---|---|---|---|
| Ge & Yu (2017) | 85.8 | 90.2 | 83.8 | 89.1 |
| $L^2$ | 79.6±0.5 | 81.4±0.2 | 83.5±0.3 | 88.2±0.2 |
| $L^2$-SP | 84.2±0.3 | 85.1±0.2 | **85.7**±0.1 | **89.6**±0.1 |
| $L^2$-SP-Fisher | 84.0±0.4 | 85.1±0.2 | **85.7**±0.1 | **89.6**±0.1 |

## 4.3 RESULTS

### 4.3.1 FINE-TUNING FROM A SIMILAR SOURCE

Table 2 displays the results of fine-tuning with $L^2$-SP and $L^2$-SP-Fisher, which are compared to the baseline of fine-tuning with $L^2$, and the state-of-the-art reference of selective joint fine-tuning (Ge & Yu 2017). Note that we use the same experimental protocol, except that Ge & Yu (2017) allow 200,000 additional images from the source problem to be used during transfer learning, whereas we did not use any.

We report the average accuracies and their standard deviations on 5 different runs. Since we use the same data and start from the same starting point, runs differ only due to the randomness of stochastic gradient descent and to the weight initialization of the last layer. Our results with $L^2$ penalty are consistent with Ge & Yu (2017).

In all experiments reported in Table 2, the worst run of fine-tuning with $L^2$-SP or $L^2$-SP-Fisher is significantly better than the best run of the standard $L^2$ fine-tuning according to classical pairwise tests at the 5% level. Furthermore, in spite of its simplicity, the worst runs of $L^2$-SP or $L^2$-SP-Fisher fine-tuning outperform the state-of-the-art results of Ge & Yu (2017) on the two Caltech 256 setups at the 5% level.

### 4.3.2 BEHAVIOR ACROSS PENALTIES, SOURCE AND TARGET DATABASES

A comprehensive view of our experimental results is given in Figure 2. Each plot corresponds to one of the four target databases listed in Table 1. The light red points mark the accuracies of transfer learning when using Places 365 as the source database, whereas the dark blue points correspond to the results obtained with ImageNet. As expected, the results of transfer learning are much better when source and target are alike: the scene classification target task MIT Indoor 67 (top left) is better transferred from the scene classification source task Places 365, whereas the object recognition target tasks benefit more from the object recognition source task ImageNet. It is however interesting to note that the trends are alike for the two source databases: all the fine-tuning strategies based on penalties using the starting point -SP as a reference perform consistently better than standard fine-tuning ($L^2$). There is thus a benefit in having an explicit bias towards the starting point, even when the target task is not too similar to the source task.

This benefit tends to be comparable for $L^2$-SP and $L^2$-SP-Fisher penalties; the strategies based on $L^1$ and Group-Lasso penalties behave rather poorly in comparison to the simple $L^2$-SP penalty. They are even less accurate than the plain $L^2$ strategy on Caltech 256 − 60 when the source problem is Places 365. We suspect that the standard stochastic gradient descent optimization algorithm that we used throughout all experiments is not well suited to these penalties: they have a discontinuity at the starting point where the optimization starts. We implemented a classical smoothing technique to avoid these discontinuities, but it did not help.

Finally, the variants using the Fisher information matrix behave like the simpler variants using a Euclidean metric on parameters. We believe that this is due to the fact that, contrary to lifelong learning, our objective does not favor solutions that retain accuracy on the source task. The metric defined by the Fisher information matrix may thus be less relevant for our actual objective that only

---

[1]The results of MIT Indoors 67 are obtained using Places 365 as source database.

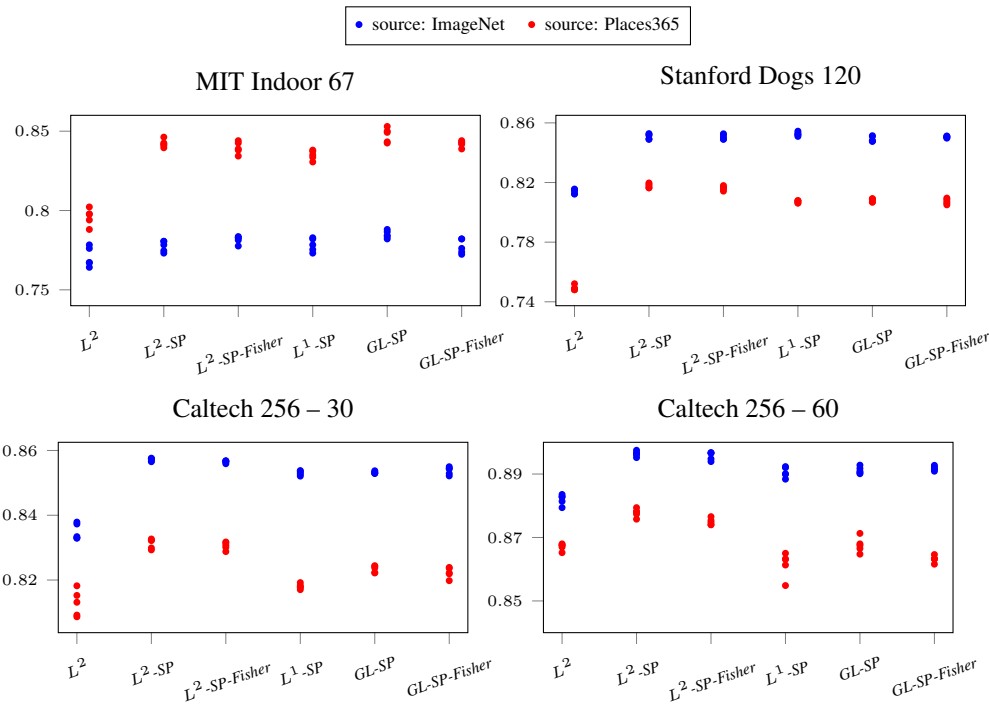

Figure 2: Classification accuracies of all tested approaches using ImageNet or Places 365 as source databases on four target databases. All -*SP* approaches outperform $L^2$. Related source task gains more performance for the target task.

Table 3: Performance drops on the source tasks for the fine-tuned models, with fine-tuning based on $L^2$, $L^2$-*SP* and $L^2$-*SP-Fisher* regularizers. The accuracies of the pre-trained models are 76.7% and 54.7% on ImageNet and Places 365 respectively.

| Source Database | Target Database | with $L^2$ | with $L^2$-*SP* | with $L^2$-*SP-Fisher* |
|---|---|---|---|---|
| ImageNet | Stanford Dogs 120 | -14.1% | -4.7% | -4.2% |
| ImageNet | Caltech 256 – 30 | -18.2% | -3.6% | -3.0% |
| ImageNet | Caltech 256 – 60 | -21.8% | -3.4% | -2.8% |
| Places 365 | MIT Indoors 67 | -24.1% | -5.3% | -4.9% |

relates to the target task. Table 3 reports the performance drop when the fine-tuned models are applied on the source task, without any retraining, but using the original classification layer instead of the classification layer learned for the target task. The performance drop is considerably larger for $L^2$ fine-tuning than for $L^2$-*SP*, and the latter is slightly improved with $L^2$-*SP-Fisher*. Hence, we confirm here that $L^2$-*SP-Fisher* is indeed a better approach in the situation of lifelong learning, where accuracies on the source tasks matter.

### 4.3.3 FINE-TUNING *vs.* FREEZING THE NETWORK

Freezing the first layers of a network during transfer learning is another way to ensure a very strong induction bias, letting less degrees of freedom to transfer learning. Figure 3 shows that this strategy, which is costly to implement if one looks for the optimal number of layers to be frozen, can improve $L^2$ fine-tuning considerably, but that it is a rather inefficient strategy for $L^2$-*SP* fine-tuning Overall, $L^2$ fine-tuning with partial freezing is still dominated by the straight $L^2$-*SP* fine-tuning. Note that $L^2$-*SP-Fisher* (not displayed) behaves similarly to $L^2$-*SP*.

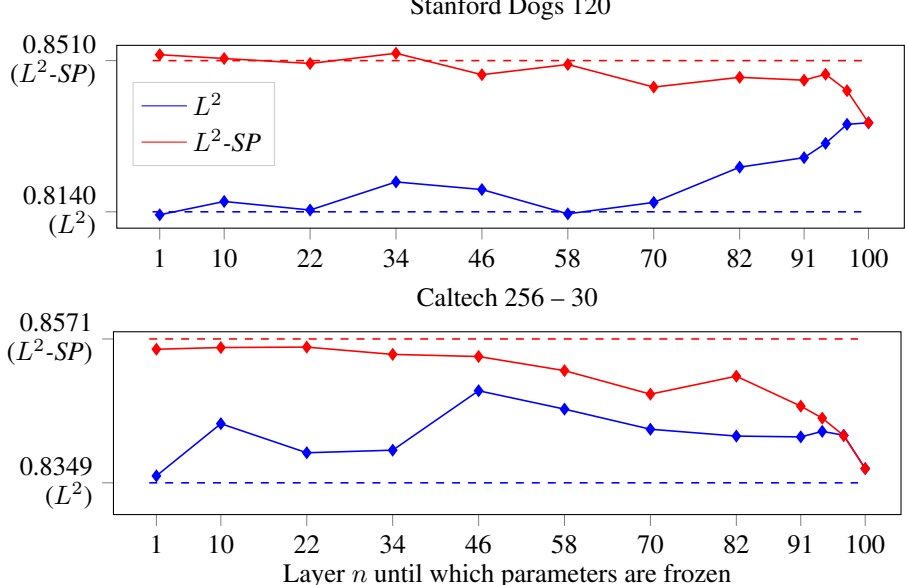

Figure 3: Classification accuracies of fine-tuning with $L^2$ and $L^2$-*SP* on Stanford Dogs 120 (top) and Caltech 256 –30 (bottom) when freezing the first $n$ layers of ResNet-101. The dashed lines represent the accuracies in Table 2, where no layers are frozen. ResNet-101 begins with one convolutional layer, then stacks 3-layer blocks. The three layers in one block are either frozen or trained altogether.

## 4.4 ANALYSIS AND DISCUSSION

Among all *-SP* methods, $L^2$-*SP* and $L^2$-*SP-Fisher* always reach a better accuracy on the target task. We expected $L^2$-*SP-Fisher* to outperform $L^2$-*SP*, since Fisher information helps in continual learning, but there is no significant difference between the two options. Since $L^2$-*SP* is simpler than $L^2$-*SP-Fisher*, we recommend the former, and we focus on the analysis of $L^2$-*SP*, although most of the analysis and the discussion would also apply to $L^2$-*SP-Fisher*.

**Theoretical Explanation**   Analytical results are very difficult in the deep learning framework. Under some (highly) simplifying assumptions, we show in Appendix A that the optimum of the regularized objective function with $L^2$-*SP* is a compromise between the optimum of the unregularized objective function and the pre-trained parameter vector, precisely an affine combination along the directions of eigenvectors of the Hessian matrix of the unregularized objective function. This contrasts with $L^2$ that leads to a compromise between the optimum of the unregularized objective function and the origin. Clearly, searching a solution around the pre-trained parameter vector is intuitively much more appealing, since it is the actual motivation for using the pre-trained parameters as the starting point of the fine-tuning process. Hence, the regularization procedures resulting in the compromise with the pre-trained parameter encode a penalty that is coherent with the original motivation.

Using $L^2$-*SP* instead of $L^2$ can also be motivated by a (still cruder) analogy with shrinkage estimation (see e.g. Lehmann & Casella 1998, chapter 5). Although it is known that shrinking toward any reference is better than raw fitting, it is also known that shrinking towards a value that is close to the "true parameters" is more effective. The notion of "true parameter" is not applicable to deep networks, but the connection with Stein shrinking effect may be inspiring by surveying the literature considering shrinkage towards other references, such as linear subspaces. In particular, it is likely that manifolds of parameters defined from the pre-trained network would provide a better reference than the single parameter value provided by the pre-trained network.

**Linear Dependence**   We complement our results by the analysis of the activations in the network, by looking at the dependence between the pre-trained and the fine-tuned activations throughout the network. Activation similarities are easier to interpret than parameter similarities and provide a

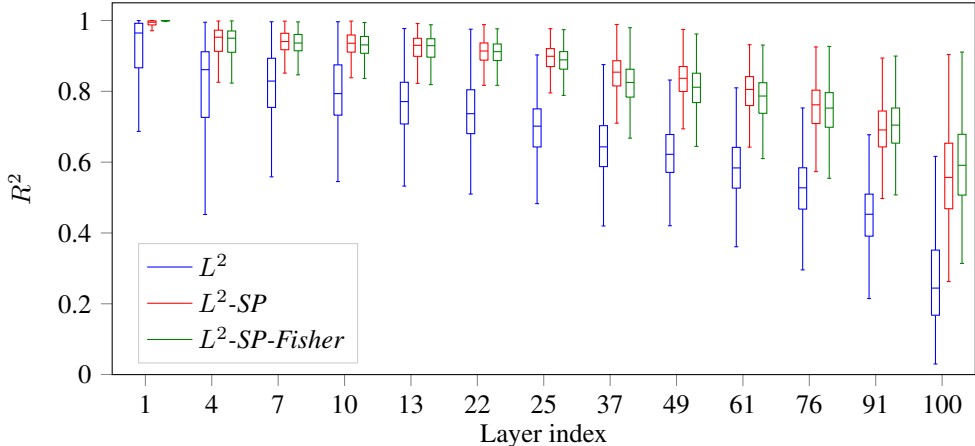

Figure 4: $R^2$ coefficients of determination with $L^2$ and $L^2$-*SP* regularizations for Stanford Dogs 120 training examples. Each boxplot summarizes the distribution of the $R^2$ coefficients of the activations after fine-tuning with respect to the activations of the pre-trained network, for all the features in one layer. ResNet-101 begins with one convolutional layer, then stacks 3-layer blocks. We display here only the $R^2$ at the first layer and at the outputs of some 3-layer blocks.

view of the network that is closer to the functional prospective we are actually pursuing. Matching individual activations makes sense, provided that the networks slightly differ before and after tuning so that few roles should be switched between feature maps. This assumption is comforted when looking at Figure 4, which displays the $R^2$ coefficients of the fine-tuned activations with respect to the original activations. We see that the $R^2$ coefficients smoothly decrease throughout the network. They eventually reach low values for $L^2$ regularization, whereas they stay quite high, around 0.6 for $L^2$-*SP*, $L^2$-*SP-Fisher* at the greatest depth. This shows that the roles of the network units is remarkably retained with $L^2$-*SP* and $L^2$-*SP-Fisher* fine-tuning, not only for the first layers of the networks, but also for the last high-level representations before classification.

**Computational Efficiency** The *-SP* penalties introduce no extra parameters, and they only increase slightly the computational burden. $L^2$-*SP* increases the number of floating point operations of ResNet-101 by less than 1%. At little computational cost, we can thus obtain 3~4% improvements in classification accuracy, and no additional cost is experienced at test time.

## 5 CONCLUSION

We proposed simple regularization techniques for inductive transfer learning, to encode an explicit bias towards the solution learned on the source task. Most of the regularizers evaluated here have been already used for other purposes, but we demonstrate their relevance for inductive transfer learning with deep convolutional networks.

We show that a simple $L^2$ penalty using the starting point as a reference, $L^2$-*SP*, is useful, even if early stopping is used. This penalty is much more effective than the standard $L^2$ penalty that is commonly used in fine-tuning. It is also more effective and simpler to implement than the strategy consisting in freezing the first layers of a network. We provide theoretical hints and strong experimental evidence showing that $L^2$-*SP* retains the memory of the features learned on the source database.

Besides, we tested the effect of more elaborate penalties, based on $L^1$ or Group-$L^1$ norms, or based on Fisher information. None of the $L^1$ or Group-$L^1$ options seem to be valuable in the context of inductive transfer learning that we considered here, and using the Fisher information with $L^2$-*SP* does not improve accuracy on the target task. Different approaches, which implement an implicit bias at the functional level, alike (Li & Hoiem 2017), remain to be tested: being based on a different principle, their value should be assessed in the framework of inductive transfer learning.

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

# A  EFFECT OF $L^2$-SP REGULARIZATION ON OPTIMIZATION

The effect of $L^2$ regularization can be analyzed by doing a quadratic approximation of the objective function around the optimum (see, e.g. Goodfellow et al. 2017, Section 7.1.1). This analysis shows that $L^2$ regularization rescales the parameters along the directions defined by the eigenvectors of the Hessian matrix. This scaling is equal to $\frac{\lambda_i}{\lambda_i + \alpha}$ for the $i$-th eigenvector of eigenvalue $\lambda_i$. A similar analysis can be used for the $L^2$-SP regularization.

We recall that $J(\boldsymbol{w})$ is the unregularized objective function, and $\tilde{J}(\boldsymbol{w}) = J(\boldsymbol{w}) + \alpha \left\| \boldsymbol{w} - \boldsymbol{w}^0 \right\|_2^2$ is the regularized objective function. Let $\boldsymbol{w}^* = \operatorname{argmin}_{\boldsymbol{w}} J(\boldsymbol{w})$ and $\tilde{\boldsymbol{w}} = \operatorname{argmin}_{\boldsymbol{w}} \tilde{J}$ be their respective minima. The quadratic approximation of $J(\boldsymbol{w}^*)$ gives

$$\mathbf{H}(\tilde{\boldsymbol{w}} - \boldsymbol{w}^*) + \alpha(\tilde{\boldsymbol{w}} - \boldsymbol{w}^0) = 0 \ , \tag{8}$$

where $\mathbf{H}$ is the Hessian matrix of $J$ w.r.t. $\boldsymbol{w}$, evaluated at $\boldsymbol{w}^*$. Since $\mathbf{H}$ is positive semidefinite, it can be decomposed as $\mathbf{H} = \mathbf{Q}\boldsymbol{\Lambda}\mathbf{Q}^T$. Applying the decomposition to Equation (8), we obtain the following relationship between $\tilde{\boldsymbol{w}}$ and $\boldsymbol{w}^*$:

$$\mathbf{Q}^T \tilde{\boldsymbol{w}} = (\boldsymbol{\Lambda} + \alpha \mathbf{I})^{-1} \boldsymbol{\Lambda} \mathbf{Q}^T \boldsymbol{w}^* + \alpha (\boldsymbol{\Lambda} + \alpha \mathbf{I})^{-1} \mathbf{Q}^T \boldsymbol{w}^0 \ . \tag{9}$$

We can see that with $L^2$-SP regularization, in the direction defined by the $i$-th eigenvector of $\mathbf{H}$, $\tilde{\boldsymbol{w}}$ is a convex combination of $\boldsymbol{w}^*$ and $\boldsymbol{w}^0$ in that direction since $\frac{\lambda_i}{\lambda_i + \alpha}$ and $\frac{\alpha}{\lambda_i + \alpha}$ sum to 1.

