# OpenReview forum: " Explicit Induction Bias for Transfer Learning with Convolutional Networks"
_ICLR.cc/2018/Conference — Reject_

### Official Review · AnonReviewer3 · 2017-11-26
**well written, needs more comparisons/analysis**

**Rating:** 6
**Confidence:** 4

**Review:**

The paper proposes an analysis on different adaptive regularization techniques for deep transfer learning.
Specifically it focuses on the use of an L2-SP condition that constraints the new parameters to be close to the
ones previously learned when solving a source task.

+ The paper is easy to read and well organized
+ The advantage of the proposed regularization against the more standard L2 regularization is clearly visible
from the experiments

- The idea per se is not new: there is a list of shallow learning methods for transfer learning based
on the same L2 regularization choice
[Cross-Domain Video Concept Detection using Adaptive SVMs, ACM Multimedia 2007]
[Learning categories from few examples with multi model knowledge transfer, PAMI 2014]
[From n to n+ 1: Multiclass transfer incremental learning, CVPR 2013]
I believe this literature should be discussed in the related work section

- It is true that the L2-SP-Fisher regularization was designed for life-long learning cases with a
fixed task, however, this solution seems to work quite well in the proposed experimental settings.
From my understanding L2-SP-Fisher can be considered the best competitor of L2-SP so I think
the paper should dedicate more space to the analysis of their difference and similarities both
from the theoretical and experimental point of view. For instance:
--  adding the L2-SP-Fisher results in table 2
--  repeating the experiments of figure 2 and figure 3 with L2-SP-Fisher

---

> ### Author Response · Authors · 2018-01-05
> **Response**
>
> Thank you for your feedback. First, we would like to inform you that we improved our results by using the original version of ResNet. The comparisons and conclusions are qualitatively identical to the previous version.
>
> (1) Thanks for the references that are now added in the related work section (J. Yang et al, T. Tommasi et al.). Although our regularizers are technically very similar to A-SVM and PMT-SVM, they have a rather different role: they act on the representation learned by the deep networks, which are equivalent to the kernels of SVM. The only weights that are not preserved by our approach are the weights in the last layer, which are the only weights that are regularized by A-SVM and PMT-SVM.
> Our aim in this paper is to promote regularizers that are technically similar to the ones used for SVM, but that are largely ignored in fine-tuning for transfer learning. This paper demonstrates that the regularization matters in transfer learning and that a simple change can improve the performance for the target task.
>
> (2) L2-SP-Fisher / L2-SP
> -- Thank you for your suggestion. We have added the L2-SP-Fisher results in Table 2.
> -- Figure 3 (Figure 2 in the previous version), the figure with frozen layers: in fact, the results with L2-SP-Fisher do not present significant differences with the ones using L2-SP. Figure 4 (Figure 3 in the previous version), the experiment on linear dependence: we have added L2-SP-Fisher results. There's no noticeable difference between L2-SP and L2-SP-Fisher but we can observe extremely high R2 in the first layer, which indicates large values of Fisher matrix and the importance of the first layer.
> -- We have also tested the performance of fine-tuned models on source tasks, just like lifelong learning problems. The -SP approaches did much better than L2. We have also observed that L2-SP-Fisher can always do better than L2-SP. This comparison has also been added in the latest version.

---

### Official Review · AnonReviewer2 · 2017-11-27
**A reasonably thorough study of regularization techniques for transfer learning through fine-tuning**

**Rating:** 6
**Confidence:** 5

**Review:**

This work addresses the scenario of fine-tuning a pre-trained network for new data/tasks and empirically studies various regularization techniques. Overall, the evaluation concludes with recommending that all layers of a network whose weights are directly transferred during fine-tuning should be regularized against the initial net with an L2 penalty during further training.

Relationship to prior work:
Regularizing a target model against a source model is not a new idea. The authors miss key connections to A-SVM [1] and PMT-SVM [2] -- two proposed transfer learning models applied to SVM weights, but otherwise very much the same as the proposed solution in this paper. Though the study here may offer new insights for deep nets, it is critical to mention prior work which also does analysis of these regularization techniques.

Significance:
As the majority of visual recognition problems are currently solved using variants of fine-tuning, if the findings reported in this paper generalize, then it could present a simple new regularization which improves the training of new models. The change is both conceptually simple and easy to implement so could be quickly integrated by many people.

Clarity and Questions:
The purpose of the paper is clear, however, some questions remain unanswered.
1) How is the regularization weight of 0.01 chosen? This is likely a critical parameter. In an experimental paper, I would expect to see a plot of performance for at least one experiment as this regularization weighting parameter is varied.
2) How does the use of L2 regularization on the last layer effect the regularization choice of other layers? What happens if you use no regularization on the last layer? L1 regularization?
3) Figure 1 is difficult to read. Please at least label the test sets on each sub-graph.
4) There seems to be some issue with the freezing experiment in Figure 2. Why does performance of L2 regularization improve as you freeze more and more layers, but is outperformed by un-freezing all.
5) Figure 3 and the discussion of linear dependence with the original model in general seems does not add much to the paper. It is clear that regularizing against the source model weights instead of 0 should result in final weights that are more similar to the initial source weights. I would rather the authors use this space to provide a deeper analysis of why this property should help performance.
6) Initializing with a source model offers a strong starting point so full from scratch learning isn’t necessary -- meaning fewer examples are needed for the continued learning (fine-tuning) phase. In a similar line of reasoning, does regularizing against the source further reduce the number of labeled points needed for fine-tuning? Can you recover L2 fine-tuning performance with fewer examples when you use L2-SP?

[1] J. Yang, R. Yan, and A. Hauptmann. Adapting svm classifiers to data with shifted distributions. In ICDM Workshops, 2007.
[2]  Y. Aytar and A. Zisserman. Tabula rasa: Model transfer for object category detection. In Proc. ICCV, 2011.

------------------
Post rebuttal
------------------
The changes made to the paper draft as well as the answers to the questions posed above have convinced me to upgrade my recommendation to a weak accept. The experiments are now clear and thorough enough to provide a convincing argument for using this regularization in deep nets. Since it is simple and well validated it should be easily adopted.

---

> ### Author Response · Authors · 2018-01-05
> **Response**
>
> Thank you for your feedback. First, we would like to inform you that we improved our results by using the original version of ResNet. The comparisons and conclusions are qualitatively identical to the previous version.
>
> Below are our answers to your questions.
>
> (0) Relationship to prior work:
> Thanks for the references on SVM that are now added in the related work section. Although our regularizers are technically very similar to A-SVM and PMT-SVM, they have a rather different role: they act on the representation learned by the deep networks, which are equivalent to the kernels of SVM. The only weights that are not preserved by our approach are the weights in the last layer, which are the only weights that are regularized by A-SVM and PMT-SVM.
>
> Our aim in this paper is to promote regularizers that are technically similar to the ones used for SVM, but that are largely ignored in fine-tuning for transfer learning. This paper demonstrates that the regularization matters in transfer learning and that a simple change can improve the performance for the target task.
>
> (1)(2) The regularization hyper-parameters are chosen in the same way as choosing other hyper-parameters, by cross validation. We have added a new figure (Figure 1) in Section 4.2 to show the sensitivity of the two regularization hyper-parameters. This figure can also respond the question (2).
> As for other regularization approaches on the last layer, we didn't observe the advantage of L1. The paper focuses on the demonstration that we can improve the performance by using a pre-trained model as reference to regularize the parameters.
>
> (3) Thank you for pointing this out. Sub-graph labels in Figure 2 (Figure 1 in the previous version) have been added.
>
> (4) Thanks for your observation about the figure with frozen layers. We corrected it by using the set of hyper-parameter values that were tested for getting the results of Table 2 (fixing no layers) throughout the experiments with frozen layers. The updated results, now in Figure 3, correspond to your expectations.
> We reproduced this experiment on Caltech 256 and found a similar pattern.
>
> (5) With Figure 4 (Figure 3 in the previous version), we verify the effect of -SP approaches on parameters by measuring the linear dependence of activations. Activation similarities are easier to interpret than parameter similarities and provide a view of the network that is closer to the functional prospective we are actually pursuing. In addition, it proves that there's some connection between preserving the parameters and preserving the activations.
> From this figure, we can also notice that comparing with L2, L2-SP always has an R2 coefficient above 0.6, which means that L2-SP is capable to keep most of the pre-trained model.
> On the other hand, we have added the results of L2-SP-Fisher in this Figure. Although there's no noticeable difference between L2-SP and L2-SP-Fisher, we can observe extremely high R2 in the first layer, which indicates large values of Fisher matrix and the importance of the first layer.
>
> (6) Totally right. Another way to observe the advantage of L2-SP is that we need fewer training examples to have the same performance with L2.

---

### Official Review · AnonReviewer1 · 2017-11-30
**limited novelty, but consistently improving fine-tuning**

**Rating:** 7
**Confidence:** 4

**Review:**

The paper addresses the problem of transfer learning in deep networks. A pretrained network on a large dataset exists, what is the best way to retrain the model on a new small dataset?
It argues that the standard regularization done in conventional fine-tuning procedures is not optimal, since it tries to get the parameters close to zero, thereby forgetting the information learnt on the larger dataset.
It proposes to have a regularization term that penalizes divergence from initialization (pretrained network) as opposed to from zero-vector. It tries different norms (L2, L1, group Lasso) as well as Fisher information matrix to avoid interfering with important nodes, and shows the effectiveness of these alternatives over the standard practice of “weight decay”.

Although the novelty of the paper is limited and have been shown for transfer learning with SVM classifiers prior to resurgence of deep learning, the reviewer is unable to find a prior work doing same regularization in deep networks. Number of datasets and experiments are moderately high, results are consistently better than standard fine-tuning and fine-tuning is a very common tool for ML practitioners in various application fields, so, I think there is benefit for transfer learning audience to be exposed to the experiments of this paper.

---- Post rebuttal
I think the paper has merits to be published. As I note above, it's taking a similar idea with transfer learning of SVM models from a decade ago to deep learning and fine-tuning. It's simple with no technical novelty but shows consistent improvement and has wide relevance.

---

> ### Author Response · Authors · 2018-01-05
> **Response**
>
> Thank you for your feedback. First, we would like to inform you that we improved our results by using the original version of ResNet. The comparisons and conclusions are qualitatively identical to the previous version.
>
> As for the novelty of the paper, we agree that there is no technical advance. However, if one agrees that the scheme we propose is very intuitive, obvious to implement, that it significantly improves accuracy and that nobody uses it, we believe that we have a (simple) message to convey to the community.

---

### Author Response · Authors · 2018-01-05
**Brief summary of the changes**

Once again, we would like to thank the reviewers for their comments and feedback. We answered to each reviewer individually and submitted a revision of the paper. Here is a brief summary of the changes:

1 Introduction: Reformulation and clarification of sentences.

2 Related Work: We have added discussions about the regularization methods used in SVM models, specifically A-SVM and PMT-SVM.

4 Experiments:
4.2: A new figure showing the sensibility of regularization hyper-parameters has been added.
4.3.1: The results of L2-SP-Fisher have been added in Table 2.
4.3.2: We have added a table showing the performance drops on the source tasks with fine-tuned models based on L2, L2-SP and L2-SP-Fisher regularizers.
4.3.3: Results of the experiment with frozen layers have been updated and the same experiment has been repeated on Caltech 256.
4.4: We have added the shrinkage estimation as another theoretical explanation.
We have also rephrased the sentences in this section.

---

### Decision · Program_Chairs · 2018-01-29
**ICLR 2018 Conference Acceptance Decision**

**Decision:**

Reject

**Comment:**

This paper addresses the question of how to regularize when starting from a pre-trained convolutional network in the context of transfer learning.  The authors propose to regularize toward the parameters of the pre-trained model and study multiple regularizers of this type.  The experiments are thorough and convincing enough.  This regularizer has been used quite a bit for shallow models (e.g. SVMs as the authors mention, but also e.g. more general MaxEnt models).  There is at least some work on regularization toward a pre-trained model also in the context of domain adaptation with deep neural networks (e.g. for speaker adaptation in speech recognition).  The only remaining novelty is the transfer learning context.  This is not a sufficiently different setting to merit a new paper on the topic.

---

> ### Author Response · Authors · 2018-02-22
> **Utility vs. Novelty**
>
> We discussed a lot about posting this comment or not... We finally would like to take the opportunity to discuss openly about the criteria for publishing papers. Novelty is obviously very important in research, because research aims at producing knowledge. However, another obvious fact is that most novelties are soon forgotten: among these, the numerous variants or combinations of existing things that do not bring much understanding or value according to whatever utility criterion one may have as a researcher or an end-user. In this regard, stressing the importance of novelty in paper evaluation is not necessarily good.
>
> Publishing should primarily a question of utility, not of novelty. Our paper asserts (convincingly If we believe the reviews), that the current strategy for transfer learning is to use a wrong baseline. This is bad if you want to get good results in applications, this is also bad if you build novel strategies motivated by improving upon a bad baseline. We therefore think that publishing this paper is useful for stopping the current practice, which provides suboptimal results, and encourages useless novel strategies that only improve upon a bad baseline.